# Multiscale Hybrid Convolutional Deep Neural Networks with Channel Attention

**DOI:** 10.3390/e24091180

**Published:** 2022-08-24

**Authors:** Hua Yang, Ming Yang, Bitao He, Tao Qin, Jing Yang

**Affiliations:** 1Electrical Engineering College, Guizhou University, Guiyang 550025, China; 2Power China Guizhou Engineering Co., Ltd., Guiyang 550001, China

**Keywords:** convolutional neural networks, feature fusion, pyramid architecture, channel attention, skip connection

## Abstract

Attention mechanisms can improve the performance of neural networks, but the recent attention networks bring a greater computational overhead while improving network performance. How to maintain model performance while reducing complexity is a hot research topic. In this paper, a lightweight Mixture Attention (MA) module is proposed to improve network performance and reduce the complexity of the model. Firstly, the MA module uses multi-branch architecture to process the input feature map in order to extract the multi-scale feature information of the input image. Secondly, in order to reduce the number of parameters, each branch uses group convolution independently, and the feature maps extracted by different branches are fused along the channel dimension. Finally, the fused feature maps are processed using the channel attention module to extract statistical information on the channels. The proposed method is efficient yet effective, e.g., the network parameters and computational cost are reduced by 9.86% and 7.83%, respectively, and the Top-1 performance is improved by 1.99% compared with ResNet50. Experimental results on common-used benchmarks, including CIFAR-10 for classification and PASCAL-VOC for object detection, demonstrate that the proposed MA outperforms the current SOTA methods significantly by achieving higher accuracy while having lower model complexity.

## 1. Introduction

The Convolutional Neural Network (CNN) has excellent feature learning ability and has been rapidly developed [1,2,3,4] in the field of computer vision, such as image classification [5,6], object recognition [7,8,9], and semantic segmentation [10,11,12]. Since the AlexNet [1] network was proposed, researchers have aproposed many other methods to improve the performance of the network. For example, the attention mechanism in natural language processing is introduced into computer vision, which can improve the performance of the network [13,14,15,16,17,18]. SENet, which obtains the channel attention weight vector by learning the interaction between channels, is the most representative. And the channel weight vector is used to scale each channel in the input feature map to highlight the useful features and suppress the useless features.

Many researchers have improved the SENet network to obtain the performance gain, but these methods easily suffer from greater computational overhead

Qin et al. [14] introduced discrete cosine transformation into the CNN and proposed a new multi-spectral channel attention mechanism. The frequency domain component index needs to be selected by three criteria and thus the model is complex. Wang et al. [15] proposed a local cross-channel interaction strategy without dimensionality reduction, which can be efficiently implemented via 1-D convolution. However, the 1-D convolution layer is difficult to model the channel information, resulting in a small network effectiveness gain. According to the input multi-scale information, Li et al. [16] used a channel attention mechanism to adaptively adjust the receptive field of each neuron in order to obtain performance gains while bringing greater model complexity. Besides the channel attention mechanism, some researchers have also introduced spatial attention mechanisms to the network, such as the SGE [17], DAN [18] and PSANet [19], to improve its performance. Nevertheless, there is a lack of channel feature modeling and this leads to more floating-point operations. Some researchers have shown that combining channel and spatial attention mechanisms can further improve network performance [12,20,21]. But these methods also bring more computational overhead, making the network efficiency worse. In order to reduce the complexity of the network, other studies have tried to simplify the structure of channel or spatial attention [22,23]. For example, Zhang et al. [22] used grouped convolution to process the input feature map, thereby reducing the parameters and computational complexity. However, this method fails to establish long-range dependency. Although the above methods can improve the performance of the network to a certain extent, they also bring greater computational overhead and higher model complexity. They can also only obtain information in the local range of the input feature map, which is ineffective at constructing large-scale dependence. Therefore, building a low-load and low-complexity network is a problem. This paper proposes a low-load and high-performance MA (Mix Attention, MA) module to solve this problem.

The proposed MA module can process the input tensor at multiple scales. Specifically, the multi-branch structure is used to aggregate the information of the input feature map. Meanwhile, each branch can effectively extract spatial information from each channel feature map at different scales by compressing the channel dimension of the input tensor. The feature maps then extracted from each branch are merged by splicing. Thus, neighbor scales of contextual features can be merged more accurately. Finally, the cross-dimensional interaction is constructed by extracting the channel attention weight of the multi-scale feature map. Softmax operation is used to recorrect the attention weight of the corresponding channel, so as to establish the long-range dependence of the channel. The MA module is used to replace the 3 × 3 convolution in the ResNet residual block to obtain an efficient mixture attention (Efficient Mixture Attention, EMA) module. A network architecture EMANet with strong feature expression ability is established by stacking the EMA modules. The main contributions of this paper are as follows:An effective MA module is proposed, which can extract multi-scale spatial information and establish channel long-range dependence. MA is a plug and play module that can be applied to various computer vision task architectures to improve the performance of the model.An effective backbone network EMANet is obtained by using the MA module instead of 3 × 3 convolution in the ResNet network, which can obtain rich feature information.Experimental results on mini-ImageNet, CIFAR-10 and PASCAL-VOC2007 datasets indicate that the proposed EMANet network achieves a distinguished performance compared with other attention networks while maintaining low complexity.

The rest of the paper is organized as follows: Section 2 introduces the channel attention mechanism and presents a pyramid compression hybrid module method. Section 3 quantitatively and qualitatively evaluates the performance of the proposed method and compares it with the baseline and existing state-of-the-art methods. Finally, Section 4 summarizes the work of this paper.

## 2. Methods

### 2.1. Channel Attention Module

The channel attention module has been widely used since it was proposed by Jie Hu, and is mainly used in various computer vision tasks. By learning correlations between channels in the input feature map, it dynamically weights each channel to enhance useful features and suppress noise. For a given feature map X∈ℝC×H×W, where *C*, *H*, *W* indicate the channel number, spatial height and width, respectively, an SE block consists of two parts: squeeze and excitation, which are used to encode the global information and calibrate the channel correlation, respectively. Generally, the global average pooling is used to compress the two-dimensional feature map into a real number, which has a global receptive field, followed by two fully connected hidden layers. The output of each fully connected layer has an activation function, which is ReLU (Rectified Linear Unit, ReLU) and Sigmoid, respectively. The linear information between channels is more effectively combined by using two fully connected layers. The average-pooling function is defined as:(1)zc=Fsqxc=1H×W∑iH−1∑jW−1xci,j
where *H*, *W* indicate the height and width of the feature map, and xci,j represents a pixel in the feature map.

The *c*-th channel attention weight can be written as:(2)Wc=Fexzc=σW1δW0zc
where δ represents the rectified linear unit ReLU operation, W0∈ℝn×n/r and W1∈ℝ(n/r)×n represent the weight of the fully connected layer, and the symbol σ represents the excitation function; usually, the channel weight vector is obtained by using the Sigmoid function, and *n* and *r* represent the number of channels and the channel decay rate, respectively. By using the excitation function, the channel weight can be allocated, so as to extract information more effectively. The channel attention weight generation process introduced above is named as the squeeze and excitation weight (SEW) module, and the schematic diagram of the SEW module is shown in Figure 1.

### 2.2. Hybrid Attention Module

This paper takes into account the hybrid idea of the ConvMixer [24] and the advantages of the multi-branch architecture of EPSANet [25]. Firstly, the input feature map is processed by multi-branch architecture, and each branch uses depthwise convolution to mix the spatial locations. Afterward, pointwise convolution is used to mix the channel locations. Large kernel convolution is used in depthwise convolution to mix remote spatial location information, so as to construct long-range dependence while obtaining larger receptive fields. Finally, a mixed attention MA module is proposed, which is composed of four parts, as shown in Figure 2. Firstly, by executing the Mixer and Concat (MC) module, the multi-scale mixed feature map is obtained. Secondly, the SEW module is executed on the multi-scale mixed feature map to obtain the channel weight vector. Thirdly, Softmax function recorrects the channel weight vector to obtain the calibrated multi-scale channel weight vector. Fourthly, the calibrated weight vector is multiplied by the corresponding channel of the multi-scale mixed feature map. And finally, a refined feature map which is richer in multi-scale feature information is obtained and used as the output.

As shown in Figure 2, in the MA module, the main operation for multi-scale mixed feature extraction is the MC module, and the overall structure of the module is shown in Figure 3. In order to extract multi-scale spatial information, the input feature map is processed in a multi-branch way, the channel dimension of the input tensor of each branch is C, and the output channel dimension is C′=C/S, where S represents the number of branches. By doing this, more abundant spatial location information can be obtained. The different spatial resolutions and depths can be generated by using multi-scale convolutional kernels in a pyramid structure. And the spatial information with different scales on each channel-wise feature map can be effectively extracted by squeezing the channel dimension of the input tensor. For each branch, it learns multi-scale mixed spatial information independently and establishes cross-dimensional interaction in a wide range. However, when the size of the convolution kernel increases, the hyperparameters also gradually increase. Therefore, in order to perform multi-scale convolution on the input tensors without increasing computational costs, grouped convolutions are heavily applied in the convolutional layers. At the same time, to select different group sizes without increasing the amount of parameters, referring to EPSANet network architecture design rules, the correlation between the multi-scale kernel size and group size can be defined as:(3)G=2K−12
where *K* represents the size of the convolution kernel and *G* is the size of the group; the effectiveness of this formula has been proved in the ablation study. For each branch, the spatial dimension of the input tensor is first compressed to extract local information, and the feature map generation function is defined as:(4)zi=BN(σ{ConvC→C′(X,ki,Gi)}), i=0,1,2,…,S−1
where the size of the *i*-th convolution kernel is ki=2×i+1+1, the size of the *i*-th group is Gi=2(ki−1)/2, σ represents the activation function GELU, and BN is the BatchNorm [26], which regularizes the tensors after activation to speed up the training of the model; zi∈ℝC′×H′×W′ represents feature maps with different scales, followed by the hybrid module. In order to mix the remote spatial location information, we increase the size of the convolution kernel to 9. Meanwhile, in order to prevent the increase of the convolution kernel size from causing more computational overhead and parameter numbers, we use deep convolution in this paper. According to research in the literature [27], if there is no identity shortcut in deepwise convolution of the large kernel, it is difficult to make it work. Therefore, a parallel shortcut branch was added for this paper. Referring to the Feed-Forward Network (FFN) design of *ViTs* architecture, we use a similar CNN-style block composed of shortcut, SoftBAN, one 1 × 1 layers and GELU to mix channel location information. Hence, each branch in the MC module is very similar to the Transformer structure. And by doing this, a larger combined receptive field can be obtained, and the cross-dimensional interaction of channels is established. In the operation of the mixing module, the spatial dimension and channel dimension of the tensor are not changed. The mixing operation function is defined as:(5)zi′=BN(σ{ConvDeptwise(zi,k=9)})+zi
(6)Fi=BN(σ{ConvPointwise(SoftBAN(zi′))})+zi′
where SoftBAN is an improvement to IEBN [28]; please check Appendix A for detailed proof.

By extracting the channel attention weight information from the multi-scale preprocessing feature map, the channel weight vectors with different scales are obtained. The channel attention weight vector can be expressed as:(7)φi=SEW(Fi), i=0,1,2,…,S−1
where φi∈ℝC′×1×1 is the attention weight, and the SEW· function obtains the attention weight from the input feature maps at a different scale. Due to the introduction of multi-branch architecture and the allocation of different convolution kernel sizes for each branch, the MA module can fuse context information at different scales, and under the holding of large kernel residual convolution, it is possible to generate better pixel-level attention for advanced semantic feature maps. In addition, in order to achieve the interaction of attention information and the fusion of cross-dimensional vectors without destroying the original channel attention weight vector, the whole channel attention weight vector is obtained by a concatenation method, as shown in Equation (8):(8)φ=Cat([φ0,φ1,φ2,…,φS−1])
where φ is a multi-scale weight attention vector.

Soft attention is used across the channel to adaptively select different spatial scales, which are guided by the channel weight vector $\varphi_{i}$. A soft weight assignment is given by:(9)$$at_{i}=Soft\max(\varphi_{i})=\frac{\exp(\varphi_{i})}{\sum_{i}^{S-1}\exp(\varphi_{i})}$$

Softmax is used to obtain multi-scale channel recalibration weights, which contain all local information in space and attention weights in channels. By doing this, the interaction between local and global attention is realized. Next, the channel attention vectors of the feature calibration are fused and spliced in a concatenation manner, so the entire channel attention vector can be expressed as:(10)$$at=Cat([at_{0},at_{1},at_{2},\ldots ,at_{S-1}])$$
where at represents the attention weight vector of the multi-scale channel after attention interaction. We multiply the recalibrated weight $at_{i}$ of the multi-scale channel attention with the feature map $F_{i}$ of the corresponding scale as:(11)$$Y_{i}=F_{i}⊗at_{i},\;i=0,1,2,\ldots ,S-1$$
where ⊗ denotes channel-wise multiplication, and $Y_{i}$ refers to the feature map weighted by the multi-scale channel attention weight vector, which has stronger feature representation and modeling capability, The concatenation operator is more efficient than the summation operator because it maintains the feature representation intact without destroying the information of the original feature map. In summary, the procedure to obtain optimized output can be written as:(12)$$Y=Cat([Y_{0},Y_{1},Y_{2},\ldots ,Y_{S-1}])$$

From the above analysis, the MA module proposed in this paper can integrate multi-scale spatial information and cross-channel attention into the blocks of each feature group. Therefore, the MA module can obtain better information interaction between local and global channel attention.

### 2.3. Network Design

The network architecture refers to the design of ResNet, as shown in Figure 4. There are two main factors to consider in choosing the residual network architecture. First, the residual network is the best performing convolutional neural network architecture in various computer vision tasks. It is meaningful to use the residual network as the backbone network to verify whether the MA structure is conducive to the mainstream CNN. Second, the residual network is conducive to the training of the network, so that the potential performance of the network is released. The overall architecture of the network is shown in Table 1. The MA module is used to replace the 3 × 3 convolutional layer in the residual network architecture, and the rest of the architecture remains unchanged. We name this network architecture EMANet.

## 3. Experimental Verification and Results Analysis

In order to verify the effectiveness of the model proposed in this paper, performance tests were performed based on mini-ImageNet, CIFAR-10 and PASCAL-VOC2007 datasets. All models were trained on NVIDIA RTX 3060Ti GPUs with 8 GB of VRAM and 16 GB of RAM, and the system was Ubuntu 20.04.4 LTS. The code and models are available at https://github.com/Xsmile-love/pytorch-emanet-master (accessed on 12 June 2022).

### 3.1. Dataset

For classification tasks, this paper uses mini-ImageNet dataset and CIFAR-10 dataset to verify the effectiveness of the proposed model. The mini-ImageNet dataset contains 100 categories, each category contains 600 images, with a total of 60,000 images; the size of each image is not fixed, the training dataset contains 48,000 images, and the validation dataset contains 12,000 images. The CIFAR10 dataset contains 10 categories of color images, each category contains 6000 images, each image size is 32 × 32; CIFAR-10 is a small dataset, a total of 60,000 images. A total of 50,000 images are used as the validation setand the rest are used as the validation set. For the object detection task, the PASCAL-VOC2007 dataset is generally used to verify the effectiveness of the model, which contains a total of 21,504 images; the training set contains 16,552 images, and the validation set contains 4952 images, with a total of 20 categories.

### 3.2. Experimental Parameter Settings

For the mini-ImageNet image classification task, the data is first augmented with random cropping, random horizontal flipping and normalization. The optimization is performed by using the stochastic gradient descent (SGD) with weight decay of 1 × 10^−4^, momentum is 0.9, cross entropy loss is used as the loss function, and the epoch is 120; the initial learning rate is set to 0.1 and is adjusted by the factor $0.1^{\mathrm{int}\left(epoch/30\right)}$, and the batch size is set to 16. For the CIFAR-10 dataset, random cropping, random horizontal flipping and normalization are used to enhance the dataset. The SGD is used with a weight decay of 0.0005, the momentum is 0.9, cross entropy loss is adopted to train all models, the learning rate is initially set as 0.1 and is adjusted by CosineAnnealingLR; the T_max and epoch are set as 200. For the object detection task, the Adam is used with a weight decay of 0.0005, StepLR is used as a learning strategy, step size is set as 1, gamma is 0.96, and the backbone network uses ImageNet 1k dataset to pre-train the weight. At the beginning of the training, the backbone network is frozen for 50 epochs. At this time, the region proposal network is trained. The learning rate in the freezing phase is 0.0001, and the batch size is set to four. All parameters are trained in the unfreezing stage, and the epoch is 100, since the memory usage is relatively large at this time, the batch size is set as two, and the learning rate in the unfreezing stage is 0.00001. 

### 3.3. Image Classification Results

We compared EMANet with the current SOTA attention methods. The evaluation metrics included both efficiency (i.e., network parameter and GFLOPs) and effectiveness (i.e., Top-1 or Top-5 accuracy). As shown in Table 2, the EMANet network proposed in this paper achieved the best accuracy on Top-1, which outperforms ResNet [4] by an above absolute 1.99%, although ResNet [4] is 10.9% larger in parameter and 8.5% larger in computation. Compared with the EPSANet [23] network, the number of parameters and floating-point operations per second was increased by 0.62 and 0.11, respectively, but the Top-1 accuracy was increased by 0.83%. Therefore, it is worth increasing these parameters and floating-point operations per second. Furthermore, with comparable or less complexity than ECANet [13], EMANet achieves above absolute 1.08% gain in performance in terms of Top-5 accuracy, which demonstrates the superiority of adaptive aggregation for a multiple branch.

In order to verify the generalization ability of the model, experiments were carried out on the CIFAR-10 dataset, and the experimental results are shown in Table 3.

As can be seen from Table 3, the EMANet network proposed in this paper achieves the optimal result of 95.61% on accuracy, which verifies the generalization ability of the MA module. It is lower than other methods except that the number of parameters and floating-point operations are 0.62 and 0.05 higher than EPSANet [25], respectively. For example, compared with the SENet [13] network, the number of parameters was reduced by 18.6% and the computational cost is reduced by 8.20%. Figure 5 visually shows that the model proposed in this paper significantly outperforms other networks. The above results show that the MA module proposed in this paper improves the performance of the network to a certain extent, and maintains fewer parameters, which proves the effectiveness of the MA module.

### 3.4. Network Visualization Results

In order to validate the effectiveness of the MA module more intuitively, nine images were sampled from the ImageNet-1k validation set, and Grad-CAM [29] was used to visualize the heatmap of layer4.2 feature maps in the EMANet network. Grad-CAM is a recently proposed visualization method, which uses the gradient to calculate the importance of spatial position in the convolution layer. Since the gradients are computed for unique classes, the Grad-CAM results can clearly demonstrate the regions that the network focuses on. By observing the regions that are considered to be very important for the prediction category, it can be seen how the network makes good use of features. For a fair comparison, heatmaps of layer4.2 feature maps in the ResNet50 network are also drawn. Figure 6 visualizes the Grad-CAM results.

It can be clearly seen from Figure 6 that the Grad-CAM mask of the network with the MA module can cover the target object region better than other methods. In other words, the network integrated with the MA module learns to take advantage of information in the target object region and aggregate features from it. Therefore, the MA module proposed in this paper can indeed enhance the expression ability of the network.

### 3.5. Object Detection Results

In order to validate the ability of the EMANet network to handle downstream tasks, pre-training was performed on the ImageNet-1k dataset, but due to the limitation of computer computing power, the remaining backbone networks listed in Table 4 were not pretrained, and the pretraining weights provided by the original author was used to train Faster-RCNN [30] on the PASCAL-VOC2007 dataset, and evaluate the bounding box Average Precision (AP) for object detection. We implemented Faster-RCNN using the MMDetection toolkit. As shown in Table 4, in the object detection task, EMANet achieved the best performance. Similar to image classification, the bounding box AP is 8.20% higher than ResNet [4], while the number of parameters and floating-point operations per second are 8.20% and 15.50% less than ResNet50, respectively. Compared with other attention networks, EMANet achieved the best performance in all metrics. It is worth noting that the EMANet network achieved 84.80% on *AP*_50_, which is 4.30%, 2.50% and 3.8% higher than SENet [13], FcaNet [14], and ECANet [15], respectively. The experimental results demonstrate that the proposed EMANet has good expression ability; when the complexity of the network is decreased, the performance is improved consistently, which proves the powerful feature expression ability of the EMANet network.

### 3.6. Ablation Study

In the pyramid architecture, a huge increase in the amount of parameters will e result from the increase in convolution kernel size. In order to extract multi-scale information from the input feature map without increasing the computational cost, this paper realized the balance between model accuracy and complexity by adjusting the convolution group size parameter, and improved the model performance by adjusting the kernel size of deep convolution kernel to mix long-distance spatial information.

Convolution group size

As shown in Table 5, this paper decreased the number of parameters and floating-point operation by adjusting group size. In the multi-branch architecture, as the size of the convolution kernel increases, the amounts of parameters will increase significantly. In order to extract multi-scale spatial information, the complexity is decreased by adjusting the group size of different branches. From the experimental results in Table 5, when the group size is (1, 4, 8, 16), a good balance can be achieved between accuracy and complexity; the experiments are performed on the mini-ImageNet dataset.

2.Mixed operation kernel size

Mixing large-range spatial location information is achieved by adjusting the kernel size in deep convolution. It can be seen from Table 5 that with the increase in kernel size, the Top-1 accuracy increases gradually, but when the kernel size is 13, the performance is significantly reduced. Therefore, the kernel size of nine is selected to mix spatial location information in this paper.

## 4. Conclusions

The purpose of the research in this paper was to improve the performance of the model with reduced complexity. To achieve the goals, we proposed a plug-and-play module, i.e., MA, which can effectively extract multi-scale spatial information and important cross-dimensional features. Therefore, it can enhance the expressiveness of the network. By leveraging an improved multi-branch architecture and channel attention mechanism, the MA module can efficiently aggregate multi-scale contextual features and image-level category information. Extensive qualitative and quantitative experiments demonstrate that the EMANet network proposed in this paper achieves the best performance across image classification and object detection tasks compared with other attention methods.

In the future, we will focus on the following tasks:The MA module will be further improved to become a lightweight plug and play module.We will use Mask-RCNN and RetinaNet detectors to verify the generalization ability of the EMANet model on the MS-COCO dataset.

## Figures and Tables

**Figure 1 entropy-24-01180-f001:**
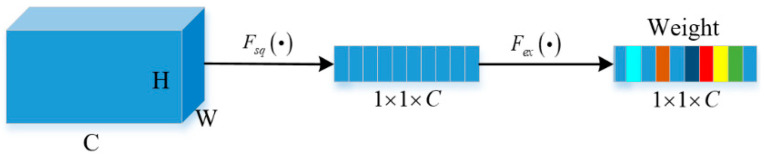
The squeeze and excitation weight module.

**Figure 2 entropy-24-01180-f002:**
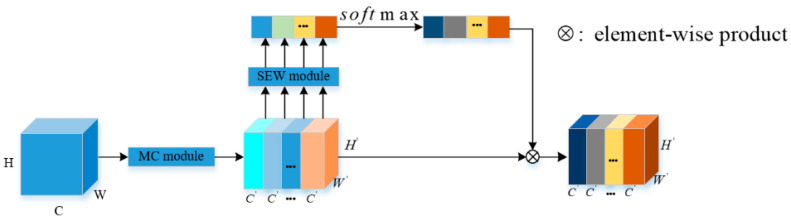
The overall architecture of the mixture attention module.

**Figure 3 entropy-24-01180-f003:**
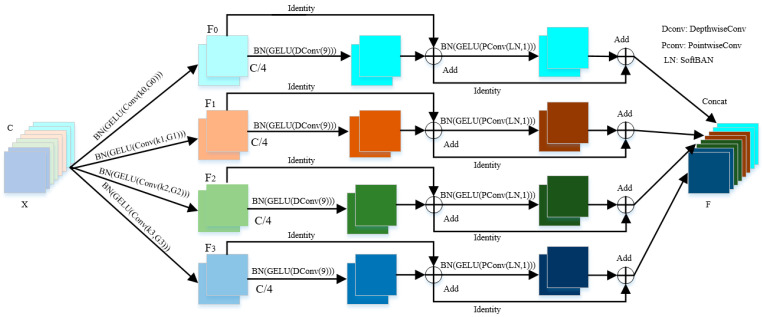
Overall architecture of MC module.

**Figure 4 entropy-24-01180-f004:**
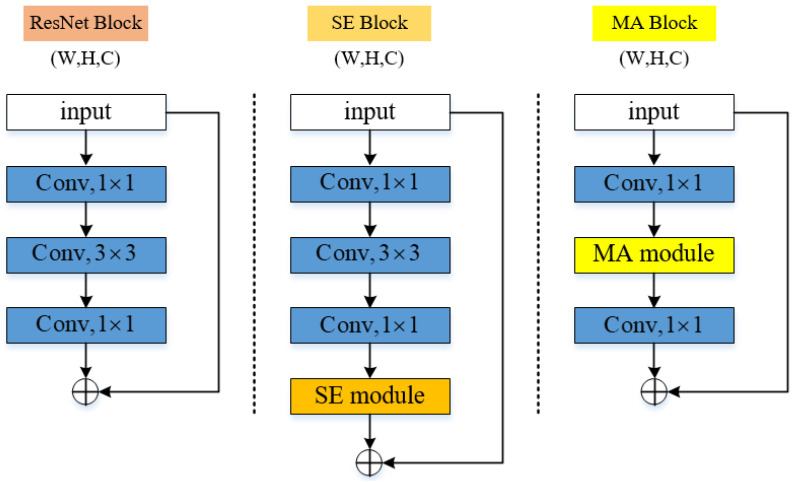
Illustration and comparison of ResNet Block, SENet Block and our proposed EMANet Block.

**Figure 5 entropy-24-01180-f005:**
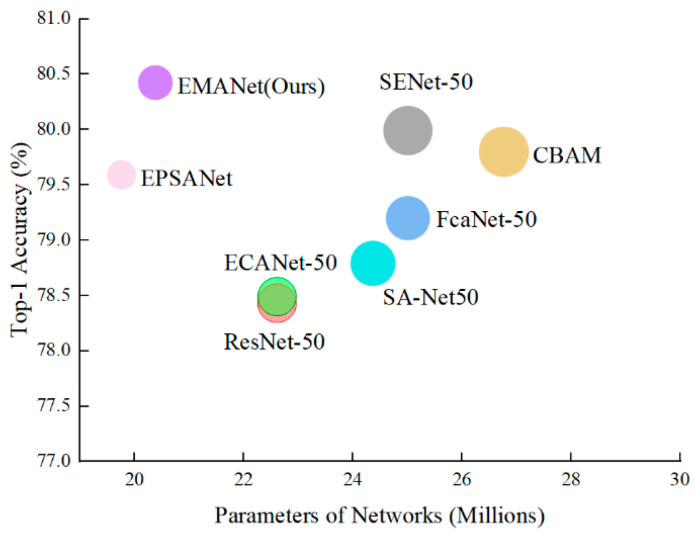
Comparisons of recently SOTA attention models on mini-ImageNet, using ResNets as backbones, in terms of accuracy, network parameters, and FLOPs. The size of circles indicates the FLOPs.

**Figure 6 entropy-24-01180-f006:**
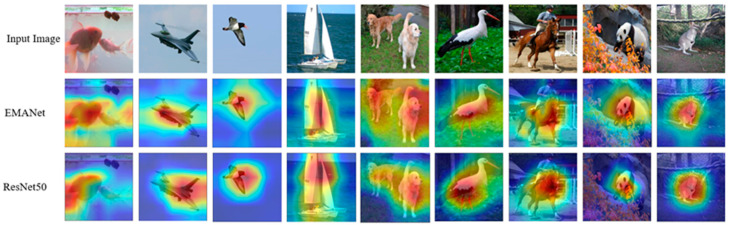
Sample visualization on ImageNet-1k val split generated by Grad-CAM. All target layers selected are “layer4.2”.

**Table 1 entropy-24-01180-t001:** Network architecture of the proposed EMANet.

Output	ResNet-50	EMANet
112×112	7×7, 64, stride 2	
56×56	3×3 max pool, stride 2	
56×56	$\left[\begin{array}{cc} 1\times 1, & 64\\ 3\times 3, & 64\\ 1\times 1, & 256 \end{array}\right]\times 3$	$\left[\begin{array}{cc} 1\times 1, & 64\\ MA, & 64\\ 1\times 1, & 256 \end{array}\right]\times 3$
28×28	$\left[\begin{array}{cc} 1\times 1, & 128\\ 3\times 3, & 128\\ 1\times 1, & 512 \end{array}\right]\times 4$	$\left[\begin{array}{cc} 1\times 1, & 128\\ MA, & 128\\ 1\times 1, & 512 \end{array}\right]\times 4$
14×14	$\left[\begin{array}{cc} 1\times 1, & 256\\ 3\times 3, & 256\\ 1\times 1, & 1024 \end{array}\right]\times 6$	$\left[\begin{array}{cc} 1\times 1, & 256\\ MA, & 256\\ 1\times 1, & 1024 \end{array}\right]\times 6$
7×7	$\left[\begin{array}{cc} 1\times 1, & 512\\ 3\times 3, & 512\\ 1\times 1, & 2048 \end{array}\right]\times 3$	$\left[\begin{array}{cc} 1\times 1, & 512\\ MA, & 512\\ 1\times 1, & 2048 \end{array}\right]\times 3$
1×1	7×7 global average pool, 1000-d fc	

**Table 2 entropy-24-01180-t002:** Comparison of various attention models on mini-ImageNet in term of network parameters, FLOPs, Top-1 and Top-5 validation accuracy.

Networks	Parameters (M)	FLOPs (G)	Top-1 (%)	Top-5 (%)
SENet [13]	25.01	3.84	79.99	94.48
ResNet [4]	22.61	3.83	78.43	93.50
FcaNet [14]	25.01	3.83	79.20	93.77
ECANet [15]	22.61	3.83	78.49	93.50
EPSANet [25]	19.76	3.37	79.59	93.75
CBAM [12]	26.77	3.84	79.80	94.52
SA-Net [22]	24.37	3.83	78.79	93.78
EMANet	20.38	3.53	80.43	94.58

**Table 3 entropy-24-01180-t003:** Performance comparison of various attention models on CIFAR-10 dataset.

Network	Parameters (M)	FLOPs (G)	Accuracy (%)
ResNet [4]	22.43	1.215	93.62
CBAM [12]	24.83	1.222	93.43
SA-Net [22]	22.43	1.216	93.79
SENet [13]	24.83	1.219	95.35
FcaNet [14]	24.83	1.217	95.49
ECANet [15]	22.43	1.217	95.35
EPSANet [25]	19.58	1.066	95.32
EMANet	20.20	1.119	95.61

**Table 4 entropy-24-01180-t004:** Object detection results of different attention methods using Faster-RCNN on PASCAL VOC2007 val dataset (*AP*_50_: AP at IoU = 0.50, *AP*_75_: AP at IoU = 0.75).

Backbone	Parameters (M)	FLOPs (G)	mAP	AP	*AP* _50_	*AP* _75_	*AP_S_*	*AP_M_*	*AP_L_*
SENet [13]	30.98	351.76	80.77	48.1	80.5	50.3	13.3	34.1	54.3
ResNet [4]	28.47	317.98	78.60	45.7	78.3	47.8	6.7	32.3	51.6
FcaNet [14]	30.98	351.63	82.53	49.4	82.3	52.6	10	34.9	55.8
ECANet [15]	28.47	351.11	81.41	48.4	81	50.9	10.2	32.9	55.1
EMANet	26.13	268.57	85.20	53.9	84.8	59	13.6	37	61.1

**Table 5 entropy-24-01180-t005:** Influence of group size and kernel size on model performance.

Group Size	Kernel Size	Parameters (M)	FLOPs (G)	Top-1 Acc (%)	Top-5 Acc (%)
(4,8,16,16)	9	16.499	2.914	78.16	93.29
(16,16,16,16)	9	15.525	2.759	78.05	93.11
(1,4,16,16)	9	19.460	3.388	79.84	93.93
(1,4,8,16)	9	20.378	3.533	80.43	94.58
(1,4,8,16)	5	20.176	3.461	72.77	90.28
(1,4,8,16)	7	20.263	3.492	78.30	93.33
(1,4,8,16)	13	20.695	3.646	79.29	93.83

## Data Availability

Not applicable.

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
