# Peer review of "Multiscale Hybrid Convolutional Deep Neural Networks with Channel Attention"

_entropy, 2022, doi:10.3390/e24091180_

Round 1

Reviewer 1 Report

An excellent paper, very well written and addressing an issue of great importance. One can really appreciate the quality of the introduction, which presents in a balanced and pedagogical way the progress since AlexNet. The Material and Methodology follows in the same vein.

Figure 3 is a bit small; you can put it across the whole width.

The MC module is based on identity, is another option possible? or not?

The results are very well presented. A classic analysis point is missing in this research. Do the different methods predict the same classes or do they potentially carry a bias. Also, would some classes not be more represented and potentially bias the results (imbalance).

Table 5 begins to look at my main question, i.e. the choice of group / kernel size, is it possible to help the scientist for these questions.

I really enjoyed reading the paper, but I didn't find a link to a tool or code (GitHub, pytorch, Keras, ...) that allows you to take what is presented and even use it in other cases. Am I missing something?

Author Response

Thank you for this comment. We have modified the relevant content you proposed

Reviewer 2 Report

Dear authors,

that work tackles several challenges that have arisen in the field of deep neural networks and the application of transformer-based architectures. Although the contributions seem important and effective, based also on the obtained results, some major issues exist that should be addressed for improving the total work and increasing the confidence of audience:

i) you repeat several times the same context through consecutive sentences (e.g., "Experimental results based on the mini-ImageNet 18 dataset show that the network parameters and computational cost of the proposed method are 19 reduced by 9.86% and 7.83%, respectively, and the Top-1 performance is improved by 1.99% 20 compared with the backbone network ResNet50. Meanwhile, the Average Precision (AP) of the 21 PASCAL-VOC2007 dataset obtained 53.90%, which is 8.20% higher than that of ResNet50. The 22 proposed MA module achieves accuracy improvement while reducing the complexity of the mod- 23 el, and the overall performance outperforms the current SOTA methods significantly"). Try to shorten some sentences and avoid such examples.

ii) The first Section includes some lengthy text dedicated to other works, without managing to balance between the achievements or the main contributions of those works and their confrontation with the proposed work. Try to clarify those references better even to audience that is not much related, highlighting the most important aspects that your work improves against them.

iii) All the figures are stacked to the text that precedes and follows them. The same issue also holds for the equation that you introduce. Please elaborate on that.

iv) Figures 5 and 6, as well as the overall experimental approach that delves into the obtained results, constitute the greatest assets of that work. However, you present some redundant information, such as the backbone and the detector columns in the corresponding tables that do not offer any information.

v) Finally, you have to attach a public repo that facilitates the reproduction of those results and can help the corresponding community to get benefited by those findings and techniques.

Author Response

Point 1: you repeat several times the same context through consecutive sentences (e.g., "Experimental results based on the mini-ImageNet dataset show that the network parameters and computational cost of the proposed method are reduced by 9.86% and 7.83%, respectively, and the Top-1 performance is improved by 1.99% compared with the backbone network ResNet50. Meanwhile, the Average Precision (AP) of the PASCAL-VOC2007 dataset obtained 53.90%, which is 8.20% higher than that of ResNet50. The proposed MA module achieves accuracy improvement while reducing the complexity of the model, and the overall performance outperforms the current SOTA methods significantly"). Try to shorten some sentences and avoid such examples.

Response 1:

Thanks you for this comment. We have shortened the sentences as follows:

The proposed method is efficient yet effective, e.g., the network parameters and computational cost are reduced by 9.86% and 7.83%, respectively, and the Top-1 performance is improved by 1.99% compared with ResNet50. Experimental results on common-used benchmarks, including CIFAR-10 for classification and PASCAL-VOC for object detection, demonstrate that the proposed MA outperforms the current SOTA methods significantly by achieving higher accuracy while having lower model complexity.

Point 2: The first Section includes some lengthy text dedicated to other works, without managing to balance between the achievements or the main contributions of those works and their confrontation with the proposed work. Try to clarify those references better even to audience that is not much related, highlighting the most important aspects that your work improves against them.

Response 2:

Thank you for this comment. We have modified the relevant content you proposed as follows:

Line 50—51, page 2: However, the one-dimensional convolution layer is difficult to model the channel in-formation, resulting in a small network effectiveness gain.

Line 56—57, page 2: Nevertheless, there is a lack of channel feature modelling and leads to more float-ing-point operations.

Line 64—65, page 2: But at the same time, it also brings more computational overhead, making the effi-ciency of the network worse.

Line 72—73, page 2: because the covariance matrix is introduced into the GSoP block, the number of net-work parameters is doubled.

Line 76—77, page 2: this method fails to establish long-range dependencies.

Point 3: All the figures are stacked to the text that precedes and follows them. The same issue also holds for the equation that you introduce. Please elaborate on that.

 Response 3:

Thank you for this comment. Following the commet, we have explained the formula in detail as follows:

In order to mix the remote spatial location information, we increase the size of the convolution kernel to 9, Meanwhile, in order to prevent the increase of the convolution kernel size from causing more computational overhead and parameter numbers, we use deep convolution in this paper. According to the discovery in reference [26], if there is no identity shortcut in deep-wise convolution of large kernel, it is difficult to work. Therefore, a parallel shortcut branch is added in this paper. Referring to the Feed-Forward Network (FFN) design of ViTs architecture, we use a similar CNN-style block composed of shortcut, SoftBAN, one 1×1 layers and GELU to mix channel location information. Hence, each branch in the MC module is very similar to the Transformer structure.

Reference 26: Ding, X.; Zhang, X.; Zhou, Y.; et al. Scaling Up Your Kernels to 31x31: Revisiting Large Kernel Design in CNNs. In Proceedings of the IEEE/CVF Conference on Computer Vision and Pattern Recognition (CVPR), New Orleans, Louisiana, USA, 19-24 June 2022.

Point 4: Figures 5 and 6, as well as the overall experimental approach that delves into the obtained results, constitute the greatest assets of that work. However, you present some redundant information, such as the backbone and the detector columns in the corresponding tables that do not offer any information.

 Response 4:

Thank you for this comment. We have deleted the Backbone column of the relevant tables and explained Detectors in this paper.

Point 5: Finally, you have to attach a public repo that facilitates the reproduction of those results and can help the corresponding community to get benefited by those findings and techniques.

Response 5:

Thanks you for this comment. We have put our codes on GitHub. GitHub linking: https://github.com/Xsmile-love/EMANet

Round 2

Reviewer 2 Report

Dear authors,

thank you for your updates. However, the first Section still includes several lengthy sentences that disrupt reader's interest. The structure of that part needs heavy elaboration. While, your code repo needs to be added into the final manuscript and of course elaborate that, since the readme file is empty and no comment exists at all into your scripts. This fact does not facilitate the reproduction of your results.

Author Response

Thank you very much for your suggestion, we have improved the relevant content.

Round 3

Reviewer 2 Report

Dear authors,

thanks for your updates. However, there is still much space for improvements. Thus, I pose here some points that need elaboration by your side:

- Readme file of the current github needs to be more insightful, following a more concrete format. Please also include the versions of the verified libraries at the end of readme. The most important issue is the shortage of comments inside the code. For example, how did you choose the parameters into the Normalize layer? Please provide further descriptions of the main.py and from which stages does that pipeline consist of.

- Table 4 again includes a redundant column that could be described into the main text

- Similarly, Table 6 can be reduced adding a column to describe the mechanism / variant of the used network, rather than adding different columns with just one matched case

- In Table 2, the unit of flops must be removed for consistency reasons with the same column in the next tables

- The presentation quality of the total manuscript still needs further improvements that should be conducted by the authors rather than the editorial team in later stages. Please be more careful with that cases.

- Figure 4 must be removed, since attaching python snippets does not keep pace with the needs of scientific manuscripts. Since you have provided the official implementation, those parts are redundant. You can describe with the appropriate notation or mathematical formulas what you actually need to highlight and discuss any technical details in the corresponding part of the experiments,

Author Response

Thank you very much for your proposal, we have modified the relevant content。

Round 4

Reviewer 2 Report

Dear authors,

you have fulfilled the most of the points that i commented.

However, the fact that you answered that comments were added into your github, is somehow strange. As I see, no comment has been added into the code. While the quality of the readme is still too poor.

You should at least try to capture that task. Its importance is crucial for a highly indexed journal.

Please elaborate on that.

Author Response

According to your advice, we have referred to these readmes in following paper on GitHub

Round 5

Reviewer 2 Report

Dear authors,

your implementation has been improved compared to the previous versions, linking some processes to other similar repos for more details or for better reproducibility.